# Capturing Pluripotency and Beyond

**DOI:** 10.3390/cells10123558

**Published:** 2021-12-16

**Authors:** Chih-Yu Yeh, Wei-Han Huang, Hung-Chi Chen, Yaa-Jyuhn James Meir

**Affiliations:** 1Department of Medicine, College of Medicine, Chang Gung University, Taoyuan 333, Taiwan; b0902057@cgu.edu.tw (C.-Y.Y.); b0902008@cgu.edu.tw (W.-H.H.); 2Limbal Stem Cell Laboratory, Department of Ophthalmology, Chang Gung Memorial Hospital, Linkou 333, Taiwan; 3Graduate Institute of Biomedical Sciences, College of Medicine, Chang Gung University, Taoyuan 333, Taiwan; 4Department of Biomedical Sciences, College of Medicine, Chang Gung University, Taoyuan 333, Taiwan

**Keywords:** pluripotency, epiblast, embryonal carcinoma (EC), embryo stem cell (ESC), epiblast stem cell (EpiSC), inner cell mass (ICM), primitive endoderm (PrE), formative cell (FC), extended/expanded potential stem cells (EPSCs)

## Abstract

During the development of a multicellular organism, the specification of different cell lineages originates in a small group of pluripotent cells, the epiblasts, formed in the preimplantation embryo. The pluripotent epiblast is protected from premature differentiation until exposure to inductive cues in strictly controlled spatially and temporally organized patterns guiding fetus formation. Epiblasts cultured in vitro are embryonic stem cells (ESCs), which recapitulate the self-renewal and lineage specification properties of their endogenous counterparts. The characteristics of totipotency, although less understood than pluripotency, are becoming clearer. Recent studies have shown that a minor ESC subpopulation exhibits expanded developmental potential beyond pluripotency, displaying a characteristic reminiscent of two-cell embryo blastomeres (2CLCs). In addition, reprogramming both mouse and human ESCs in defined media can produce expanded/extended pluripotent stem cells (EPSCs) similar to but different from 2CLCs. Further, the molecular roadmaps driving the transition of various potency states have been clarified. These recent key findings will allow us to understand eutherian mammalian development by comparing the underlying differences between potency network components during development. Using the mouse as a paradigm and recent progress in human PSCs, we review the epiblast’s identity acquisition during embryogenesis and their ESC counterparts regarding their pluripotent fates and beyond.

## 1. Introduction

Genome plasticity creates cellular diversity by giving rise to different cell fates, which was essential for multicellularity during evolution. In eutherian mammals, embryonic development starts from a fertilized egg (i.e., zygote), which can produce an entire organism and functional extraembryonic structures for metabolic exchanges during embryogenesis. The cell state that enables a single cell to establish extraembryonic tissue (i.e., placenta) in addition to the embryo proper is known as totipotency [1,2,3]. Following the developmental trajectory, zygotes gradually lose their potency during the differentiation process. Before implantation, the embryo contains two distinct structures, the inner cell mass (ICM) and trophectoderm, including three differentiated cell types: epiblast, hypoblast (primitive endoderm; PrE), and trophoblast. The epiblast will further develop into the embryo proper and germ cells, whereas the trophoblast and hypoblast will develop into the extraembryonic structures of the placenta and yolk sac, respectively. At this preimplantation stage, the epiblast reaches a state known as pluripotency, with the ability to form germ layers and the germline at the onset of the postimplantation stage [4]. Therefore, unlike a zygote, individual epiblasts in the pluripotent state no longer have the capacity to form an organism because they have lost the ability to form extraembryonic structures [4]. The feature of divergent developmental tracks in embryonic and extraembryonic entities in early embryos constitutes an important mechanism that arose during placental mammals’ evolution. This disparate organization of fetal development is essential and effective, since allocating extraembryonic tissue development at the onset of embryogenesis facilitates the eventual separation of the fetus and placenta at birth without complicating the design of the body plan [5,6].

Following the specification trajectory, cell potency is gradually restricted but not precipitately lost. For example, preimplantation epiblasts retain the capacity to form germ layers and the germline, and these lineage precursors can be observed only after implantation. This preservation of cell potency during early embryogenesis allows researchers to capture their unique characteristics in vitro, recapitulating their endogenous counterparts. In vitro cultured epiblasts, namely, embryonic stem cells (ESCs), derived from the microdissected inner cell mass (ICM) of late preimplantation blastocysts, retain the pluripotent cell state of epiblasts and are poised for lineage specification. The ICM in the preimplantation blastocyst constitutes a couple of dozen cells that are transient in the strictly controlled developmental course. In contrast, many ES cells can be cultured in vitro and maintain their pluripotency characteristics indefinitely. Thus, ESCs capture the preimplantation embryo stage with a particular regulatory genetic network governing the pluripotent state in a self-renewing fashion.

In this vein, the unbiased capacity to develop germ layers and the germline offers an unparalleled opportunity to unravel the mechanism of cell lineage specification and its clinical applications. With advanced genome manipulation tools, mouse disease models have been created to reveal the etiology of pathologies, which seamlessly translates into medicine in clinical settings. Here, we revisit the developmental history of embryonic stem cells and delineate recent breakthroughs regarding the complicated regulatory mechanisms of cell potency establishment from an integrated perspective. We focus on these issues by using mouse preimplantation embryogenesis as a paradigm and comparing it with its human counterpart, including the maintenance and transition mechanisms between different pluripotent states and the totipotent phase. In addition, key unresolved issues and critical questions highlighted in current stem cell biology are discussed.

## 2. Mouse Preimplantation Embryonic Development

The onset of embryogenesis begins with a zygote. Following successive division, the zygote forms a multicellular blastomere [7,8] (Figure 1A). The earlier produced blastomeres do not exhibit differences in cell fate until the fifth mitotic division. Therefore, the individual blastomere is considered to be totipotent before the occurrence of the fifth cell division. With the fifth cell division, the blastocyst is formed in the developing embryo. Although an individual blastomere in the four- or eight-cell stage can contribute to embryonic and extraembryonic tissues by aggregating with the host embryo in the same stage, only a single individual two-cell blastomere has the ability to independently develop into an entire organism. This difference could be due to the reliance of late-stage blastomeres on interactions with others during normal development [9]. Thus, strictly speaking, only the two-cell stage embryo has the totipotency capacity [10,11].

Establishing various cell lineages is an imperative feature of a multicellular organism. Transducing the positional information from the embryo architecture, which gradually shapes cell potency and specifies cell identity during embryogenesis, is the key to diversifying cell lineages. In the mouse, the first lineage decision follows a cellular compaction event, an E-cadherin-mediated intercellular interaction, in the late eight-cell stage (Figure 1B). Architecturally, blastomere compaction provides an essential spatial cue that determines subsequent cell fate according to the cell’s location [22]. Typically, the outer cells of the embryo are predisposed to form the extraembryonic lineage, whereas the inner cells tend to become the ICM. In addition to the variance in the spatial allocation and polarity of cells, differential expression profiles between extraembryonic (trophectoderm; TE) and embryonic (ICM) lineages are evident. Recently, Kinisu and colleagues identified Klf5 plays an essential role in such biopotential blastomeres through directly inducing ICM and TE specification genes [23,24]. The ICM maintains the expression of Oct4 and Sox2, which collaboratively target other pluripotency genes. For example, FGF4 and Nanog are directly downstream of Oct4 and Sox2 [25,26]. By contrast, Id2 is the earliest marker observed in the extraembryonic trophectoderm [27]. In addition to these different cell markers, cell adhesion molecules, gap junctions, and the Hippo-YAP/TAZ pathway also play significant roles in the first cell lineage establishment (Figure 1B) [12,13,14]. Despite the distinct molecular features occurring during early blastocyst formation, the cell fates of outer and inner blastomeres at the 16-cell stage are still changeable if they are reintroduced to the embryo earlier than the 16-cell stage. Thus, the plasticity of cell fate determination is one of the unique characteristics of the blastomere at this stage [28,29]. However, this plasticity is eventually lost after the fifth division because the second lineage decision is initiated, which results in the emergence of epiblasts and hypoblasts (primitive endoderm; PrE) within the ICM cell population [29,30,31].

The early mouse ICM, which forms after approximately the fifth cell division (E3.25–E3.50), consists of two cell populations: the one expressing Klf2, Sox2, Nanog, and Fgf4 will create the epiblast, whereas the other, displaying early PrE-markers that include Gata6 and Fgfr2, will become the hypoblast [21,27,31]. The lineage segregation of epi- and hypoblasts from the ICM follows the Fgf-mediated lineage specification pathway. Following the self-enhancement feedback loop to re-enforce their respective cell fate decisions, the increased expression of FGF4 and FGFR2 in epi- and hypoblasts, respectively, confirms their distinct cell identities [27] (Figure 1C). Although the upstream signaling via FGF4/FGFR2 signaling to result in a mutually exclusive process of NANOG and GATA6 to define the respective epi- and hypoblast is well recognized [18,32], how the exact nature of such bifurcation fate establishment remains unclear. Lines of evidence have suggested that the downstream effector of FGF4/FGFR2 axis, the ERK signaling output, determines epi- and hypoblast cell fate [27,33,34,35,36]. Recently, Pokrass and colleagues observed the cell-cycle-dependent ERK outputs count for epi- and hypoblast fate specification in mouse preimplantation embryos. That said, a pulsed ERK inhibition and rapid ERK activation at the mitotic exit in ICM blastomeres define epiblasts and hypoblasts, respectively [37]. Such bifurcation occurs stochastically but not via a hardwired fate designation, adding another layer of cell fate plasticity by creating heterogeneity in ICM during multicellular organisms’ development. Such flexibility in cell fate determination sheds light on how multicellular organisms ensure their success in embryo development. In that vein, having a sufficient ICM cell number would be a pre-requisite to hosting such a stochastic mechanism consistent with a critical size limitation governing the proper formation of the embryo proper in the blastocyst [38].

Further, the following activation of the Erk pathway, independent of Fgf4 and LIF, drives the differentiation of epiblasts [39,40,41]. After reaching the implantation stage, both cell lineages are irreversibly committed to their fates [29,42]. Thus, pluripotent epiblasts will continue to form the embryo proper, whereas the hypoblasts form the primitive endoderm, an extraembryonic structure of the future yolk sac. Harvesting and maintaining epiblasts from E4.5 embryos in vitro creates the well-known embryonic stem cells (ESCs). However, historically, the establishment of ESCs was not so straightforward. Instead, it began with the conceptual establishment of pluripotency in embryonal carcinoma cells (ECs) derived from spontaneous testicular teratomas.

## 3. Capturing Pluripotency

### 3.1. A Brief Historical Overview of the Unraveling of Pluripotency

Pluripotent cells have the ability to produce diverse differentiated cell types. Pluripotency was originally observed in spontaneous testicular teratomas in the 129 inbred mouse strain [43]. These testicular teratoma-derived cells were named embryonal carcinoma (EC) cells, in which a single cell can generate teratocarcinoma containing a range of differentiated cell types [44]. Interestingly, extraembryonic endoderm-like cells are always the first cell type to form once ECs are induced to differentiate, recapitulating the differentiation path of lineage specification seen in the ICM of mouse preimplantation blastocysts [45]. Subsequently, ECs were shown to be chimera-competent but lacking germline transmission because of their malignancy. However, malignant ECs can be guided to develop into typical, non-malignant differentiated cell types through induction [46]. Therefore, ECs behave like normal early embryonic cells in terms of their further differentiation via the typical development route. The shared characteristics between ECs and embryonic stem cells (ESCs) were therefore evident. The first ESC was isolated from the ICM of a delayed mouse blastocyst from the 129 inbred mouse strain [47].

Following the track of preimplantation development, the epiblasts, as mentioned above, will expand to increase their number, since a critical size limitation governs the proper formation of the embryo proper in the blastocyst [38]. Imperatively, these pluripotent epiblasts are sensitized to physiological stimuli and, in turn, modulate cell lineage commitment. As embryonic stem cells (ESCs) captured in their pluripotent state in vitro are derived from late preimplantation epiblasts (E4.5 stage), the development of ESCs—the in vitro counterparts of epiblasts—pauses under culture conditions [47,48]. However, like the progressively developing epiblasts in the preimplantation embryo, ESCs still preserve their differentiation capability. Therefore, ESCs share most features with the epiblasts seen in diapause embryos [49].

In the past few decades, ESCs have been observed to display heterogeneity within a population [50,51,52,53]. This phenomenon results from fluctuating pluripotency-associated factors in a dynamic equilibrium. Thus, even in a pluripotent state, individual cells do not necessarily have equivalent potency, reflecting their distinctive morphologies. Additionally, the stage-specific epiblast and the culture conditions greatly influence the differentiation capacity of ESCs [54,55]. After delineating the pluripotent transcriptional network and establishing a new culture system, more potency states were observed in ESCs. Surprisingly, extraembryonic markers were limited to a small fraction of the ESC population and epiblasts in the cultured blastocyst [56,57]. Thus, the phenomenon of cell fate fluctuation further extends beyond pluripotency, reaching a totipotency-like state, which has a different transcriptional circuitry from pluripotency. The transition path of fluctuating ESC fates can resemble the developmental continuum, if not wholly recapitulate preimplantation embryogenesis. This part of the mechanism is further described in the metastability section below.

Although the intriguing phenomena mentioned above have been partly clarified due to rapid advancements in research, it is interesting to note that initially, ESCs could only be established from certain mouse strains (i.e., permissive strains). Subsequently, revealing the pluripotent regulatory network and genetic variants removed this ESC-harvesting restriction in different genetic backgrounds and has further shaped our current understanding of the essence of pluripotency.

### 3.2. Permissive and Nonpermissive Mouse Strains for Establishing Pluripotency

Early efforts to establish the first ESC line were strictly dependent on the 129 inbred mouse strain. Intriguingly, ESCs could not be derived from all Mus musculus strains, despite using the same culture conditions as those used for the 129 strain-derived ESCs and despite their ability to interbreed and produce fertile offspring. The permissive strain for ESC production included 129 and its substrains and the commonly used inbred mouse strain C57BL/6. By contrast, ESCs from nonpermissive mouse strains, as the name suggests, such as CBA, NOD, and DBA, were either extremely difficult or impossible to obtain [58]. This discrepancy was initially attributed to the high incidence of spontaneous testicular teratocarcinoma observed in the 129 strain, which is not the case in other strains. However, further genetic studies from Threadgill’s group showed that nonpermissiveness is a recessive phenotype. Nonpermissive genomes fail to complement nonpermissive strains, indicating that the same group of genes or pathways governs the establishment and maintenance of ESC pluripotency in vitro [59]. It was not until the finding of intrinsic self-renewal and pluripotency properties of ESCs that the phenomenon of strain-dependent ESC harvesting was overcome in nonpermissive strains.

Evidence confirmed that the instability of nonpermissive strains’ genetic backgrounds leads to the failure to capture their pre-embryonic epiblasts under the conventional serum + LIF culture condition. Later, it was further shown that the NOD strain, a nonpermissive strain that is highly resistant to ESC derivation, favors differentiation into the postimplantation epiblast-like state (the primed state of pluripotency) [60,61,62,63]. Even though the postimplantation epiblast is also characterized by pluripotency, it is in a state of priming for lineage specification. Therefore, the potency of such epiblasts was named the primed state to distinguish it from the naïve/ground state observed in preimplantation epiblasts. Accordingly, preventing the preimplantation epiblast from differentiating into its primed pluripotency state is essential for ground state establishment, regardless of its background. Although there are clues regarding the distinctive variations in establishing ESCs from different mouse strains, the molecular basis of the difficulty deriving ESCs from nonpermissive strains remains elusive. Batlle-Morera et al. suggested that different mouse genetic backgrounds may have strain-specific variations in Erk signaling [64], the activation of which is independent of Fgf4 and LIF autocrine signaling in the embryo [39,40,41,65,66]. Because Erk activity is associated with ESC differentiation but not proliferation, inhibiting Erk prevents ESC differentiation and does not affect its self-renewal [39,63]. Additional inhibition of glycogen synthase kinase-3 (GSK3) further promotes the differentiation arrest of ESCs [67]. Therefore, by concomitantly inhibiting MEK/ERK and GSK3 pathways by culturing cells in 2i/LIF (2iL) medium, the ground state of pluripotent ESCs can be maintained irrespective of the mouse genetic background, and it has even been successful in rats [63,68,69].

Notably, Mus musculus is the only known species requiring leukemia inhibitory factor (LIF) for maintaining ESC pluripotency in vitro. In contrast, other species are LIF-independent and form the postimplantation-like epiblast (the primed state). Thus, identifying ground (naïve) and primed pluripotent states has shaped the concept of pluripotency and advanced our current understanding of early lineage commitment [70]. Recently two laboratories independently identified the drivers of strain-dependent regulatory elements using different but complementary genetics approaches to unravel the genetic variation underlying the pluripotent mESC establishment in the defined condition [71,72]. Both groups identified the Lifr expression level, and the Wnt pathway varies to impact pluripotent cell states. Such differences are also present in human pluripotency counterparts and are consistent with variegation findings in Wnt signaling outputs affecting the differentiation of human-induced pluripotent stem cells [73,74]. Thus, using a finite amount of gene numbers in a genome, multicellular organisms adopt single nucleotide polymorphism (SNP) to create diverse genetic backgrounds through having different signaling outputs. Presumably, adding another layer of diversity in the same species shall endow the fitness during evolution. Figure 2 shows mouse embryogenesis at the preimplantation stage and the potency states corresponding to epiblasts and ESCs isolated in different stages.

### 3.3. Naïve Pluripotency

Mouse embryonic stem cells (mESCs) can be established from cells in different preimplantation stages, including single blastomeres [75,76,77,78]. However, the most efficient source of mESCs is from the ICM of the mid- and late blastocyst (E4.5); i.e., all epiblasts in this stage can become ESCs [21]. The resultant ESCs share similar characteristics and closely resemble the late preimplantation epiblast, indicating that they are retained at a typical developmental stage characterized by self-renewal in addition to the ability to govern lineage specification [79]. Although ESCs are in vitro cultured cells, the self-renewal property is unlikely to be an artifact because it can also be observed in the epiblast counterpart of the diapause embryo [49]. Further evidence showed that self-renewal occurs in the brief period between the late preimplantation embryo and implantation, associated with the concomitant repression of ERK activity [21].

In contrast to the developing epiblast, which is only transiently in a state of pluripotency and is highly susceptible to extrinsic signals, cultured ESCs are capable of being perennially maintained without differentiating with their pluripotency captured in vitro. Although ESCs are paused in the pluripotent state, they still preserve their sensitivity to signals and can thus proceed with normal differentiation given the proper signal inputs. This property can be demonstrated by transferring ESCs to the stage-equivalent embryo and obtaining chimeric offspring.

In this captured naïve pluripotent state, several unique features are observed, including cell morphology, expression profile, and epigenomic configuration. Regarding ESC morphology, these cells exhibit a high nuclear-cytoplasmic ratio, forming dome-shaped clones with a reflective boundary. ESCs express a set of pluripotent markers, including Oct4, Sox2, Klf2, Nanog, and Fgf4. There is a correlation between the naïve epiblasts in embryos and ESCs in vitro regarding the suppression of DNA methylation in lineage-specific genes and the transient activation of the paternally inherited X chromosome [80,81,82]. However, these properties of epiblasts dissipate after implantation. A new epigenetic landscape will take over to prime cells for germ layer formation and lineage specification during the subsequent gastrulation event.

### 3.4. Primed Pluripotency

Immediately after implantation, the expanded epiblasts rearrange from a loosely unstructured spherical-shaped cell mass to a cup-shaped single columnar epithelium layer susceptible to lineage-specifying signals from adjacent extraembryonic tissues. The unique transcriptional profile of the postimplantation epiblast is established by repressing the naïve pluripotency network (e.g., Tfcp2/1, Rex1, Klf2, Klf4, and Tbx3) and promoting the primed state circuitry, such as Pou3f1, Otx2, and FGF5 [21,54,55,83]. The driving forces that determine this cell fate change and coordinate the structure formation mainly rely upon FGF signaling and the Erk cascade. However, neither signal directly causes lineage specification. Instead, they elevate ESC sensitivity to further inductive signals. Conceivably, the lack or obstruction of FGF4/Erk signaling in the ESC or epiblast impairs lineage commitment, allowing the maintenance of an uncommitted state [84]. Together, these signal inputs and the upregulation of Nodal and Acvr2b induce the subsequent germ layer specification during the process of gastrulation [21].

Following the transition of the signaling network and transcription regulatory circuitry from pre- to early postimplantation development, the epiblast stem cells (EpiSCs) isolated from this stage resemble their embryonic counterparts of the anterior primitive streak [54,55,85]. These changes also reflect their distinct culture conditions [54,55]. In line with these observations, FGF and Activin A are required for EpiSCs to maintain their primed pluripotent state, whereas these factors will induce ESC differentiation. Conceivably, EpiSCs cannot survive in the 2i-based medium, wherein the Erk cascade is blocked [63,67,86,87]. In contrast to ESCs, EpiSCs coexpress pluripotent and lineage-specific genes and, therefore, are poised for lineage specification. This lineage-priming pluripotent state is recognized as a primed state. Although EpiSCs can form teratomas when introduced into adult mice, they display little or no competency to form chimeras.

Several efforts have been dedicated to converting mEpiSCs to the mESC naïve state by expressing exogenous Klf2 and Nanog [88]. Recently, Stuart et al. demonstrated that transcriptionally and mechanistically distinct routes could lead to cell identity transition [89]. Through the overexpression of Klf2, Stat3, and Esrrb individually in mEpiSCs, all transgene-manipulated cells can reach a naïve state using distinct pathways; for example, induction by Klf2 proceeds via a mesoderm-like form, whereas overexpression of Stat3 adopts an early ICM-like state. Thus, cellular identity is now considered a multidimensional attractor state, allowing different intermediate states between cell identities’ exact start and end [89]. In this case, distinct routes all converge on Oct4 expression as a ‘transition factor’ to permit the identity establishment of naïve cells via various intermediate states [89]. Despite different intermediate states existing between well-recognized states, it is known that the provided niche at the intended endpoint will play an essential role in attracting to that defined state (i.e., the naïve state culture medium in this case). Besides the specific fate maintaining, such a niche establishment also engages cells’ self-renewal in that state. Similarly, adopting different routes in a reprogramming process can also be appreciated from the point of view of chemical- and transgene-based reprogramming, where the chemical reprogramming takes the XEN-like state path before reaching naïve pluripotency [90,91,92].

### 3.5. Formative Pluripotency

The paradigm of mESCs delineates the pluripotent state of cells possessing the ability to generate all germ layers and the germline, giving rise to pups with fertility [93]. In contrast, mEpiSCs are refractory to chimera formation, and if developed, viable chimeric pups do not show germline transmission [94]. Thus, along the temporal developmental continuum, the timing of germline fate determination falls between naïve and primed states. Interestingly, the ESC itself is unable to follow the instructions of transcription-factor- and growth-factor-based cues toward germ cell fate [94,95]. Instead, mESCs must first exit the naïve state of pluripotency and become epiblast-like cells (EpiLCs) to respond to germline induction cues [94]. EpiLCs are distinct from ESCs and EpiSCs in terms of their transcriptome profiles and competency for germ cell induction. Therefore, EpiLCs represent a unique germ cell lineage-specified state acquired following the differentiation trajectory of ESCs during development, but they are not in a pluripotent state. Accordingly, between naïve and primed pluripotent states, another pluripotent state must exist that is sensitized to induction cues from its surroundings and formatively makes the decision to be a germ cell or somatic lineage [96].

This year, Kinoshita et al. identified and maintained such a unique state between naïve and primed pluripotency in both mouse and human cells, which can be obtained through either stage-specific embryo isolation or ESC induction. This newly captured pluripotent state possessing the above-mentioned developmental characteristics was designated as the formative state [97]. Genome-wide transcriptome analyses indicate that the mouse formative state was similar to the pre-gastrulation epiblast (E5.5–6.0). Formative cells decrease the expression of the naïve ESC transcription network while inducing certain transcription factors, such as Otx2, Pou3f1, and Dnmt3a/3b de novo methyltransferases [21,98,99,100,101]. As expected, formative cells are competent for primordial germ cell (PGC) induction and somatic lineage specification. However, this process differs from the refractory PGC induction observed in ESCs and EpiSCs [94,102,103,104]. As formative cells resemble the pre-gastrulation E5.5–6.0 epiblast, they can contribute to blastocyst chimeras, although with lower efficiency than ESCs, whereas this is not usually observed in EpiSCs [55,105,106]. Notably, the formative cells display a greater propensity to form the mesendoderm and neural lineage in mice. However, it may take 24 h longer to reach the same efficiency as EpiSCs during neuroectoderm induction [97]. As pre-streak epiblast-derived cells, formative cells have pre-streak epiblast features, such as germline competency, primitive streak formation, and neural lineage specification. Further genome-wide transcriptome analyses indicated that formative cells are more related to EpiLCs and deviate from EpiSCs [97].

In the human counterpart, the functional criteria of formative cells are somewhat different from those of mouse formative cells. Such differences may be because conventional hPSCs share similar features with EpiSCs, but they are not identical [107,108]. For example, primordial germ cell-like cells (PGCLCs) can be induced from hPSCs [109,110]. These different properties of primed pluripotent stem cells between rodents and humans could be attributed to their architectural differences in postimplantation development. In mice, the epiblast forms an egg cylinder composed of simple columnar cells, whereas human and non-human primate embryos develop a bilaminar disc [111,112]. Although whether such structural differences lead epiblasts to adopt different cell fates is debatable, the differential induction signals received from the surrounding tissues cannot be the same [112,113,114]. Primordial germ cells (PGCs) originate from the posterior epiblasts in pre-gastrulated postimplantation embryos in mice. Although human PGCs are currently considered to be created from epiblasts, similar to their mouse counterparts, a recent study on non-human primate embryos suggested that the nascent amnion could also be a site of PGC specification. Thus, in addition to the noticeable structural embryonic differences, the transcriptional regulatory network warrants factoring into the divergence in the cell fate determination. However, due to the inaccessibility of human postimplantation embryos, more non-human primate studies are necessary to define the difference between humans and rodents in the future. Together, the formative pluripotent state represents a transition phase between naïve ESCs and prime EpiSCs. It corresponds to the pre-streak epiblast during early postimplantation and has full competency to respond to somatic and germline lineage formation signals [96,97].

### 3.6. The Counterbalance of Pluripotent Factors Defines the Pluripotent State

The pluripotent state in the epiblast of the preimplantation embryo is transient, whereas the ESC counterpart of pluripotency can be perennially maintained. Thus, the following question arises: What are the basal components and their resultant regulatory circuitry that dictate the captured pluripotent state without further proceeding to differentiation? As they are similar to master regulators, it was first proposed that pluripotent factors, the trinity comprising Oct4, Sox2, and Nanog, prevent ESCs from differentiating by repressing lineage-specific genes to reach an intrinsically stable ground state both in vitro and in vivo [84,115,116,117,118,119]. Thus, intuitively, overexpression of individual pluripotent factors should maintain the pluripotent state of ESCs by arresting their differentiation. However, the results of these studies were not consistent with these predictions. For example, overexpression of Oct4, Sox2, and Nanog individually prompts mesodermal, neuroectodermal, and mesendodermal differentiation, respectively [120,121,122,123]. Likewise, additional pluripotency factors also share the same characteristics when overexpressed [124,125]. Thus, these observations are different from the expectation that pluripotent factors serve as lineage-specific blockades as previously proposed.

To interpret the results of overexpressing pluripotent factors mentioned above, Loh and Lim proposed an alternative model in which pluripotent factors function as lineage specifiers [126]. Instead of maintaining an intrinsically stable ground state through the downregulation of lineage specifiers, the naïve pluripotent state is more likely defined by transcriptional competition between pluripotent factors that specify one of the developmental germ layers while mutually inhibiting others. As these pluripotent factors govern lineage specification by establishing a precarious counterbalance, stochastically gaining or waning individual pluripotent factors may tip toward one end and, in turn, trigger the differentiation process. Extrinsic and self-activated autocrine signals may adopt this differentiation scenario by eliciting this intrinsically unstable pluripotent state toward the designated lineage. For example, LIF signaling upregulates Klf4, Tbx3, Sox2, and Nanog [127], whereas TGFb signaling promotes Oct4 expression [128]. Similarly, FGF4, acting as an auto-inductive stimulus to drive ESCs toward lineage specification, was observed to be directly regulated by Oct4 and Sox2 [40]. Thus, carefully modulating the extrinsic signal inputs driving the expression levels of pluripotent factors allows naïve pluripotency to be maintained indefinitely in vitro [126].

### 3.7. The Human ESC

The first human pluripotent stem cells (hPSCs) were successfully isolated and maintained by James Thomson [129]. Interestingly, hPSCs are different from mouse ESCs in terms of their morphology, culture conditions, and transcription factor circuitry, despite being isolated from the same embryonic stage as their mouse counterparts. Notably, these prominent features are similar to those of the mouse EpiSCs derived from the primitive streak stage (the late epiblast of postimplantation embryos), which are not responsive to LIF, instead of relying upon FGF and Activin [54,55,85,130]. Thus, hPSCs have been considered to be the mouse EpiSC counterparts; they are in a primed state but not the naïve state of ESCs. These distinguishing characteristics were initially regarded as species-specific features, where the hPSC lacks the naïve transcriptional network seen in the mouse. Further studies, however, verified that these differences are due to different developmental phases [131].

As mentioned above, mouse EpiSCs can be converted to ESC-like by overexpressing defined transcription factors (i.e., Klf2, Nanog, and STAT3) in a 2iL medium [132,133,134]. Furthermore, the observation of mouse EpiSCs grown on feeders showed a marginal ability to undergo ESC conversion when transferred to a 2iL medium [135,136,137]. Presumably, the promiscuous signal inputs from serum and feeders promote a shift in pluripotency from the primed to naïve state through the formative phase [96]. Thus, these previous studies established a paradigm for the possibility of human ground-state ESCs. The first attempts to produce naïve hPSCs created an unstable cell state with an ESC-like morphology [138,139]. Further efforts have included formulating a complicated culture medium for in vitro maintenance of these cells [140,141,142]. However, these cells still required FGF, TGFb, and serum replacement to induce the conventional hPSC propagation without evidently activating their ground state transcription network, as seen in their mouse counterparts. Motivated by the different conceptualizations for achieving naïve pluripotency from Hanna’s and Janeisch’s groups, Chen and colleagues reinforced the STAT3 signaling pathway in hESCs by overexpressing the hormone-dependent STAT3-ER in human PSCs. Under the condition of 2i/LIF and tamoxifen, hPSCs entered a naïve-like state and were named TL2i cells [143]. Furthermore, TL2i cells can self-renew over a longer term in FGF2- and feeder-free conditions, with a gene expression profile and epigenetic configuration similar to naïve mouse PSCs. Although Chen’s approach allows hESCs to settle independently of FGF2 and self-renew in a LIF-dependent state, TL2i cells have no second X chromosome reactivation [143]. Later, Takashima et al. successfully produced naïve-like hPSCs by adopting a similar approach and expressing exogenous Klf2 and Nanog to convert mEPiSCs into mESCs [88]. Naïve-like hPSCs could then be derived from conventional hPSC (e.g., H1, H9, and Shef6) and hiPSC (human induced pluripotent stem cell) lines (e.g., fibroblast- and adipocyte-derived). These naïve-like hPSCs share the characteristics seen in mESCs, including (1) shifting metabolic uses (glycolysis in conventional hPSCs vs. oxidative phosphorylation in naïve-like hPSCs), (2) reorganizing the global DNA methylation pattern (hypermethylation in conventional hPSCs vs. hypomethylation in naïve-like hPSCs), (3) suppressing the X chromosome by epigenetically reducing foci of H3K27me3 in naïve-like hPSCs, (4) forming a functional pluripotency circuitry (e.g., TFCP2L1 and KLF4 are critical factors for naïve-like hPSCs but dispensable for conventional hPSCs), and (5) suppressing Erk signaling [141]. Although Takashima’s approach led to the generation of naïve-like hPSCs, several arguments remain to be addressed. For example, unlike the mouse, there is no endogenous Klf2 expression in the ICM of human preimplantation embryos. Thus, overexpression of Klf2 and Nanog may rewire the endogenous pathway to promote a shift in potency from the primed phase to the naïve state in hPSCs. A similar scenario may also occur in the re-enforcement of Stat3 expression [134,143]. Even though the shutdown of Klf2 and Nanog transgene expression remains sustainable for the maintenance of naïve-like hPSCs, the residual expression of Klf2 and Nanog cannot be ignored. Collectively, most currently generated naïve-like hPSCs are incapable of tolerating MEK/ERK inhibition, reactivating the X chromosome, using the distal Oct4 enhancer element for OCT4 expression, and maintaining the adequate DNA methylation level in the imprinted loci [144,145,146,147]. However, further advances under conditions free of transgenes, histone deacetylase inhibitors, and feeders will ultimately lead to bona fide naïve human pluripotency.

Recently, Hanna’s lab recently tackled this issue aiming to study the signaling circuitry of naïve human pluripotency. By synergistic inhibition of WNT/beta-CATENIN, protein kinase C (PKC), and SRC signaling, they generated teratoma-competent naïve human PSCs. Moreover, these naïve hPSCs can further differentiate into trophoblast stem cells and extraembryonic endodermal cells in vitro, representing the most stringent naïve hPSCs created to date [148]. Similarly, Guo and colleagues found that naive hPSC is not for lineage restriction in human pluripotent stem cells (hPSCs), unlike the mouse counterpart; the naïve hPSCs can differentiate into trophectoderm, whereas the primed hPSCs only can form amnion [149]. These observations were in the in vitro culture hPSC and the human blastocyst, while such epiblast plasticity in hPSC maintains until the formative transition, providing blastocyst regeneration opportunity. However, one cannot learn whether such a lineage-unrestrictive nature in the naïve hPSC also occurs in vivo, and whether the extended lineage limitation is only observed in specific genetic backgrounds or is a unique property in naïve primate PSC. It would be interesting to thoroughly compare all the naïve-like ESCs mentioned above with human and primate ICMs.

Together, species differences in the wiring of ground-state transcription factors do exist between mice and humans. Such variance may result in an unstable naïve pluripotent circuitry that drives the ephemeral developmental progression during preimplantation in human embryogenesis. Consistent with this, the diapause phenomenon is not observed in the preimplantation human embryo. Thus, the unique nature of naïve hPSCs that makes their in vitro capture tantalizing until now represents a divergence between species during evolutionary adaptation.

## 4. Metastability

Even though ESCs resemble preimplantation embryo-derived epiblasts, the plasticity of the ESC epigenome allows them to respond to environmental signals and results in divergent developmental regulations. Thus, the regulatory circuitry and fate status of ESCs differ depending upon their culture conditions. For example, ESC morphology and gene expression are notably distinctive in the presence and absence of feeders. Under feeder-free conditions (only supplied with serum and LIF), ESCs display flattened and heterogeneous morphologies, in stark contrast to feeder conditions, wherein homogenous clusters of small, tightly packed cells are formed, with a reflective boundary around each clone. Furthermore, although both culture conditions allow ESCs to express Oct4 and Sox2, heterogeneity has been observed in several pluripotent factors, including Nanog, Rex1, Esrrb, Klf4, and Stella [50,51,52,53]. Notably, this heterogeneous expression pattern dynamically oscillates, resulting in a variegated expression pattern of these pluripotent factors.

The stochastic expression of certain pluripotent factors implies that these ESCs may transiently situate between different states in a cycling-in-and-out manner. The fluctuating expression pattern of pluripotent factors in cells was recognized as a metastable phenomenon [50]. The first observed pluripotent factor with a fluctuating expression is Nanog; the ESC shuttles between low-Nanog (priming for differentiation) and high-Nanog (pluripotent state) states, corresponding to a reversible priming process of differentiation. Further evidence showed that these oscillating pluripotent factors were all downregulated at the onset of ESC differentiation and remained low in postimplantation EpiSCs, along with the upregulation of lineage-specific genes [79]. Therefore, the fluctuating expression of these genes mirrors the unstable cell status, shuttling between naïve and primed pluripotent states under these environmental conditions [53]. Therefore, feeder-free ESCs in serum and LIF comprise both self-renewing and differentiation-priming populations [53].

In line with the aforementioned observations, the term metastability was further extended to describe the transient cell state changes within the cell population, which may resemble the developmental continuum, if not the same process [60]. Accordingly, the naïve and primed pluripotent states can be considered a metastable equilibrium influenced by their microenvironment. In addition, the transition between states can be achieved by exogenous factors with suitable culture conditions [60]. In addition to these pluripotent factors with a variegated expression pattern, Canham et al. surprisingly reported that Hex, the extraembryonic endoderm (PrE) marker, also sporadically appeared in mESC culture. Hex-positive cells coexpress some pluripotent factors but are interconvertible with Nanog expression [150]. Similarly, Morgani et al. also reported the heterogeneity of Hex expression in the epiblast of the harvested preimplantation mouse embryo (E3.5~E4.5) in a 2i embryo culture [57]. This observation demonstrated that the 2i condition prevents ESCs from differentiation priming but not from adopting an extraembryonic cell fate. As the capacity to produce both extraembryonic and embryonic tissues is defined as a totipotency-like state, these observations further stretch the metastable spectrum of ESCs from the primed state to the totipotency-like phase.

According to the above-mentioned proposed concept of the precarious balance required to maintain pluripotency, the metastable phenomenon can also be explained from the model of cross-repressive lineage specifiers. Accordingly, the mechanistic view of metastability is attributed to the dynamic outputs of mutually antagonistic lineage specifiers weighted and driven by surrounding input signals [151,152]. Although changing cell fates through the fluctuation of factors may follow differentiation paths, the dynamic heterogeneity of ESCs may not be strictly in line with the developmental trajectories in the embryo. For example, Flipczyk et al. reported that the periodical timing between the gain and wane of factor expression might extend over several cell cycles, inconsistent with typical embryonic development [153]. Furthermore, the metastability phenomenon is only an epiphenomenon that is likely derived from in vitro signal inputs, since the reversible transition of cell fate lacks in vivo evidence. Nevertheless, the metastability observed in a population of ESCs offers an opportunity to study the relationship between ESC potencies and the differentiation trajectory in vitro.

The change of epigenetic configuration is considered the potential mechanism of metastability, allowing the cell fate shift and creating population heterogeneity. A two-way relationship between transcription factors and chromatin structure has been proposed to function in concert for achieving such epigenetic modifications [154,155]. Initially, such epigenetic change can be obtained from the symmetry-breaking of a replicative DNA during cell proliferation [156]. The hemimethylated strand from newly synthesized daughter DNA may not be maintained by the UHRF1/DNMT1 complex in time. It then creates a brief window to diversify the epigenetic configuration by the signal inputs from its surroundings. Similarly, the dynamic bivalent chromatin structure adds another factor in cell fate fluctuation. Such a heterogeneous DNA methylation pattern endows two daughter cells adopting different cell fates.

## 5. Capturing Totipotency

The metastable spectrum of ESCs extends to pluripotency and beyond; i.e., it possesses the capacity to form the Hex+ extraembryonic structure in addition to the embryo proper itself [57,150]. In 2012, Macfarlan et al. further reported an intriguing metastable phenomenon in a rare ESC subpopulation with a fluctuating expression of endogenous retrovirus transcripts that were only identified in the two-cell embryo stage (2C) [56]. The endogenous retrovirus belongs to the MERVL subfamily (murine endogenous retrovirus with leucine tRNA primer) reviewed in [157,158,159,160].

This transient expression of these blastomere-stage genes indicates that the cell state is beyond pluripotency and may shift toward totipotency at that specific moment. This dynamic phenomenon of transiently cycling-in-and-out between pluripotent and 2C-like totipotent states was observed in nearly all ESCs [56]. Notably, these oscillating 2C transcripts contain endogenous retroviral long terminal repeat elements located in their upstream regulatory sequences. Such foreign elements governing early cell fate are uniquely observed in placental mammals [161]. Interestingly, approximately 25% of Oct4 and Nanog recognition sites in mouse and human pluripotent cells are primarily associated with MERVL elements [162]. However, only 5% of these target genes are orthologous between species. Thus, recruiting retroviral LTR as a regulatory element at the onset of embryogenesis is a mechanism common to placental mammals, arising during their evolution. However, different species using the same regulatory process on species-specific target genes can be considered one of the pieces of molecular evidence on the speciation-diversifying strategy [162]. MERVL transcript expression is initiated during zygotic genome activation (ZGA), reaches its peak expression at the two-cell stage in mice and 4~8-cell stages in humans, and is subsequently broadly silenced in most differentiated cell lineages.

### 5.1. Mechanism

Profiling the two-cell-stage embryos and analyzing their epigenetic configuration revealed that the unique property of totipotency is distinctive from pluripotency. The prominent features of totipotency include the following: (1) expression of two-cell-embryo-specific transcripts [56], (2) downregulation of pluripotency genes [56], (3) increased histone mobility [21], (4) dispersed chromocenters [163], and (5) increased developmental capacity [56]. Regarding the molecular paths of shifting the pluripotent cell fate to a totipotent state and returning it to its original pluripotent state, Zhang’s group dissected the process of fate transitions by using single-cell RNA-seq (scRNAseq) to profile the intermediate states. Furthermore, to facilitate the enrichment of the two-cell-like state, they used doxycycline-mediated induction of the Dux gene in ESCs. Intriguingly, they concluded that the gain and wane of totipotency from pluripotent states use different routes in this potency-transitioning roundtrip [164,165] (Figure 3).

### 5.2. The Pluripotency-to-2C-Like Transition

In the course of the pluripotent-to-2C-like transition, upregulation of two-cell-embryo-specific elements, followed by downregulation of pluripotent genes, was observed, where Myc and Dnmt1 served as sequential roadblocks en route [164]. Thus, Myc is the first encountered barrier halting the downregulation of pluripotent genes, whereas Dnmt1 impedes two-cell-embryo-specific gene expression at the second step by maintaining DNA methylation. Further evidence supporting this notion was derived from the finding that knocking down both Dnmt1 and Myc led to more of a 2C-like state than knocking down either gene alone [164]. Similarly, in an independent study, Rodriguez-Terrones et al. identified a Zscan4+ intermediate state by analyzing the expression of 93 genes and obtained similar results [166]. Although both groups adopted different approaches, the common conclusion in both studies was that two-cell state entry requires the cells to first exit the pluripotent state.

### 5.3. The 2C-Like-to-Pluripotency Transition

In 2020, Fu et al. continued their line of research by studying the reverse path, i.e., from a two-cell-like state back to pluripotency [165]. As mentioned above, the route taken is not a direct path reversal of pluripotency to a two-cell-like state. Instead, it is characterized by a two-wave process of upregulating pluripotent genes and silencing specific two-cell state genes. Accordingly, the early-activated pluripotent genes (the first step in proceeding from the two-cell-like state to the intermediate phase), including Sox2, Esrrb, Arid5b, and Zfp42, participate in embryonic development and stem cell maintenance. The late-activated pluripotent genes (the second phase in shifting from the intermediate stage to the pluripotent state) include signaling, cytoskeleton, and cell junction regulatory genes. The reversal of the two-cell-like state is partly reminiscent of the transcriptional dynamics of preimplantation development via a rapid decrease in Dux mRNA and elevation of pluripotent gene expression [165]. Although the two-cell-like state may facilitate our understanding of preimplantation development, one should be aware that the 2C-like transition system remains different from in vivo embryonic development [161,167,168].

### 5.4. The 2C-Like Network

As the metastability of the molecular roadmap between two-cell-like and pluripotent states has been uncovered, as mentioned above, it is essential to identify which totipotent factors participate and how the totipotency network forms at the onset of life. Although it remains elusive, recent studies have made a leap of progress regarding these topics. Therefore, unraveling these intriguing issues will facilitate the capture of totipotency in vitro.

After fertilization, zygotic genome activation (ZGA) is initiated and peaks at the two-cell stage, characterized by the activation of 2C-specific genes, including Zscan4, Dux, and endogenous retroviruses. The interaction between the maternal factor NELFA (negative elongation factor A) and Top2a at the two-cell state allows Dux expression [169]. Alternatively, Dux expression can also be directly driven by the DPPA2/4 heterodimer [170,171] (Figure 4B right panel). Dux encodes a homeodomain-containing transcription factor and functions as a 2C key regulator. In addition to activating Dux, the DPPA2/4 heterodimer also directly expresses LINE-1, serving as a scaffold to recruit Nucleolin and KAP1 (TRIM28) to repress Dux expression [171,172] (Figure 4B left panel). This LINE-1-mediated feedback mechanism explains the observation that DUX only increases its expression at the two-cell stage and degrades afterward. This strict temporal control of DUX expression ensures that early embryo development continues advancing to the next pluripotent state and, therefore, initiates lineage specification.

As a transcription factor, DUX directly activates the expression of Zscan4d, MERVL, and miR-344 (Figure 4A). In turn, miR-344 represses the ZMYM2 complex, which negatively regulates GATA2 and MERVL-related 2C genes [173]. Eif1a (eukaryotic translation initiation factor 1A), downstream of Zscan4d, is a global translation machinery inhibitor that halts the expression of pluripotent genes (i.e., Oct4, Sox2, PRDM14, and AP2γ) and chromatin modifiers (i.e., DNMTs) but does not affect their transcripts [174]. A transient lack of DNMT proteins in these cells, particularly Dnmt1, causes temporary genome-wide DNA hypomethylation. The downregulation of Zscan4d and Eif1a indicates the departure from the two-cell stage and entry into a pluripotent state. The simultaneous translation of these pluripotent genes and Dnmt1, released from the Eif1a-mediated translation blockade, ensures the successful relay between toti- and pluripotency in terms of their transcriptional networks and epigenetic configuration establishment.

Evidently, Dux plays a pivotal role at the two-cell-like stage. However, embryos that lose Dux (Dux−/−) still produce viable pups in sub-Mendelian proportions, indicating the existence of differences between in vitro and in vivo situations [161,168]. Such discrepancies may be reasoned from the previously revealed fact that Dux can access only about 25% of the two-cell-like specific regulatory regions. Thus, other factors may also redundantly establish a two-cell-like totipotent state [175]. Identifying these missing elements will help to complete the jigsaw puzzle of the totipotent landscape. Recent studies showed that in addition to Dux, the deficiency of other 2C marker genes, including Smarca5, Nelfa, Dppa2, and Dppa4, also did not impair the expected development in the mouse, at least in the preimplantation stage of ICM and TE specification. That said, unknown factors exist governing the bipartite cell fate [168,176,177,178]. Kinisu and colleagues identified that Klf5 plays an essential role in biopotential blastomeres [23]. Although the upregulation of endogenous virus (ERV) has always served as an indication for cell fate reaching beyond pluripotency, the role of ERV elements as drivers or passengers remains elusive. Yet, compared with the Expanded pluripotent stem cells (EPSCs), which produce both embryonic and extraembryonic potency as addressed in the following section, EPCs do not induce the expression of MERVL [179,180]. From this point of view, the expression of ERV is more likely a consequence of zygote gene action but not a driven force.

### 5.5. Expanded/Extended Pluripotent Stem Cell (EPSC)

As previously mentioned, the two-cell-like state is a spontaneous process observed in a small fraction of mESCs, and shares a significant part of the transcriptome with two-cell-stage embryos [56,163,166,181]. Notably, although 2CLCs emerge from ESCs, they do not exist during development and cannot be derived from embryos. Further, because of the failure to capture and maintain the observed transient cycle-in-and-out phenomenon in vitro, there is no evidence of the self-renewal property to date. Although 2CLCs contribute to embryonic and extraembryonic lineages via blastocyst injection and facilitate nuclear transfer efficiency [56,163,182], they have, however, not been demonstrated to contribute to the morula-stage embryo, as its name suggests. Notably, the 2CLC metastable phenomenon has only been found in mESCs.

Two independent studies recently established another cell state, named extended/expanded potential stem cells (EPSCs), from mouse and human PSCs [179,180]. Although 2CLCs and EPSCs share the feature of “expanded cell fate” compared to naïve PSCs, their transcriptome profiles are genuinely different. Both 2CLCs and EPSCs have the extraembryonic tissue-forming capacity in chimeras. However, EPSCs express extraembryonic markers in culture, whereas 2CLCs do not. Most importantly, EPSCs are reprogrammed cells that arise from a complicated medium formulation (i.e., LCDM medium) to allow maintenance and self-propagation, in contrast to the ephemeral state of 2CLCs [179,180]. Thus, the rare 2CLCs pose a challenge requiring further biochemical and molecular investigations. Moreover, the recent establishments of human naïve PSCs, as aforementioned, can form trophectoderm in both in vitro culture and human blastocyst [148,149]. Therefore, addressing the essential differences between human naïve PSCs and EPSCs regarding their regulatory networks will be crucial to unravelling the plasticity of human genetic circuitry in the preimplantation embryo.

Finally, totipotent-like cells are often used to refer to both 2CLCs and EPSCs. However, according to the strict definition of totipotency, a single cell type will develop into an organism if transferred into an empty zona pellucida and implanted in a foster mother. Intuitively, this is unlikely to be the case for 2CLCs and EPSCs, since their cell contents and sizes are far different from those of a two-cell blastomere. Therefore, unraveling the differences between the two-cell blastomere, 2CLC, and EPSC, at the molecular level, warrants further investigation.

## 6. Conclusions

With the advent of regenerative medicine, understanding the characteristics of cell stemness is essential for its clinical applications. Since the first mouse ESC line was established, the past 40 years of intensive studies on the rodent model have led us into new eras. Using mouse ESCs as a paradigm, the mystery of their human PSC counterparts was rapidly unraveled. In addition, as more pluripotent states have been captured, the developmental continuum is increasingly being represented in vitro. Further recent progress in primate ESCs has contributed to filling the gap and distinguishing differences between rodents and humans, leading to the urgent need to build human (or primate) genetic circuitry.

Although metastability is an epiphenomenon observed in the in vitro system, the reversible cycling-in-and-out between different pluripotent phases of ESCs represents their pluripotent landscape’s high sensitivity and plasticity. Precariously balanced lineage specifiers in the pluripotent ESC network allow a rapid response to signal inputs for dealing with versatile situations during development. Since genomic plasticity allows ESC fate to stretch far beyond pluripotency and reach the totipotent-like state, endogenous retroviral elements’ role in regulating totipotent genes was recognized. Are these endogenous retroelement expressions sufficient to drive the beginning of life or just a consequence of ZGA? What are the other factors besides DUX required to host the totipotent state? How do the maternal factors wire into the zygotic circuitry to complete the circle of life? All of these intriguing questions warrant further study. Undoubtedly, unraveling the mechanism of genome plasticity, delineating the cell fate transition, and determining the route to the designated cell fate shall pave the way for future cell-based therapy.

As genome plasticity seemingly endows the flexibility of cell fate conversion if the cell is accidentally lost, it also provides the driving forces for species diversification during evolution. Then the question is, how does such plasticity generate diversity from a fixed number of genes within a genome? Recent findings have revealed nature’s ways, including (1) SNP grades the signaling outputs [71,72,74], (2) stochastic gene expression within a defined group of cells safeguards and, in turn, warrants the bifurcation fate decision [37,183], (3) distinct routes allow cell fate shifting between two defined cell fates [89], and (4) differential signaling–regulatory wiring extends cell potency [148,149]. Although the findings mentioned above were adopted in different phases of cell fate determination, they exhibited the diversified mechanisms successfully used in contemporary multicellular organisms for their fitness. Additionally, the findings also warrant that diversification in the present species will occur continuously.

## Figures and Tables

**Figure 1 cells-10-03558-f001:**
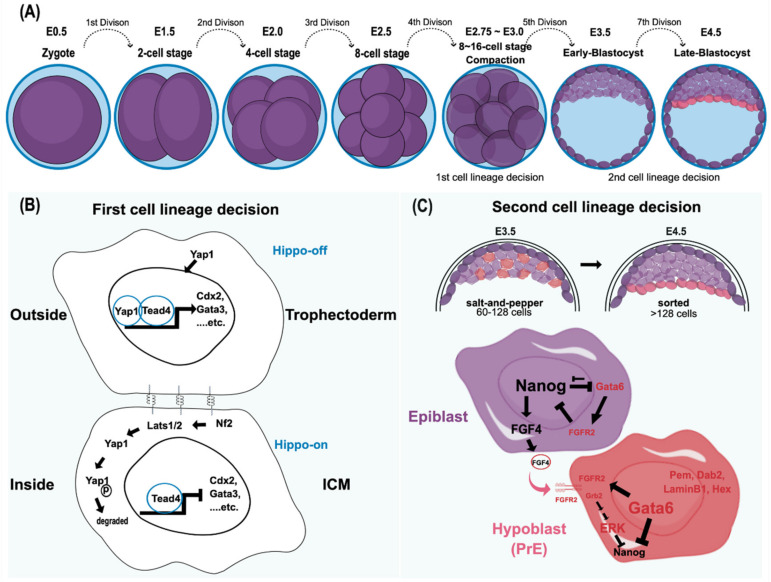
Mouse preimplantation embryo development and its cell lineage decisions. (**A**) The cell stages leading to blastomere compaction are shown. Although both cell fates are still reversible at this stage, individual blastomeres that remain outside are highly likely to become trophectoderm, whereas the cells located inside produce ICM. (**B**) This cell fate adaptation largely depends on the Hippo pathway [12,13,14]. In the Hippo-off state, Yap1 translocates into the nucleus and functions together with Tead4 to express trophectoderm markers, including Cdx2 and Gata3. By contrast, in the Hippo-on state, Yap1 is degraded through the phosphorylation of Lats1/2, and the cells adopt the ICM cell fate [13,14]. Cell–cell interactions through gap junctions, focal adhesion kinase, and integrins play a significant role in the first cell lineage decision. In essence, the decision to form trophectoderm versus ICM is based on the intercellular spatial organization with the activation of specific transcriptional programs, such as master regulators and other signals. (**C**) The second lineage specification, reorganizing and differentiating into epiblast and hypoblast (primitive endoderm; PrE) in ICM, starts around E3.5, where the mid-blastocyst is formed. Both cell allocation and lineage marker expression in ICM determine lineage specification. Both Gata6 (the PrE marker) and Nanog (the epiblast marker) are coexpressed in all ICM cells, from the morula to the early blastocyst. Later, these markers are mutually exclusive with an antagonistic relationship to reinforce this divergent lineage event [15,16,17,18,19,20]. In addition, the FGF/MAPK signaling pathway is also believed to participate in this process. The epiblast precursors secrete FGF4, whereas the PrE precursors express its receptor, FGFR2, to repress Nanog expression through the Grb2-dependent pathway [21]. The PrE markers include Gata6, Gata4, Hex, Sox17, Sox7, Dab2, and LaminInB1. In contrast, because the epiblast precursors lack FGFR2 to respond to FGF autocrine signaling, Nanog expression is preserved to define its pluripotency. Initially, the epiblasts and PrEs are intermingled in a salt-and-pepper fashion at early E3.5. Eventually, both epiblasts and PrEs are sorted into the adjacent layer by cell movement. Ⓟ indicates the phosphate group.

**Figure 2 cells-10-03558-f002:**
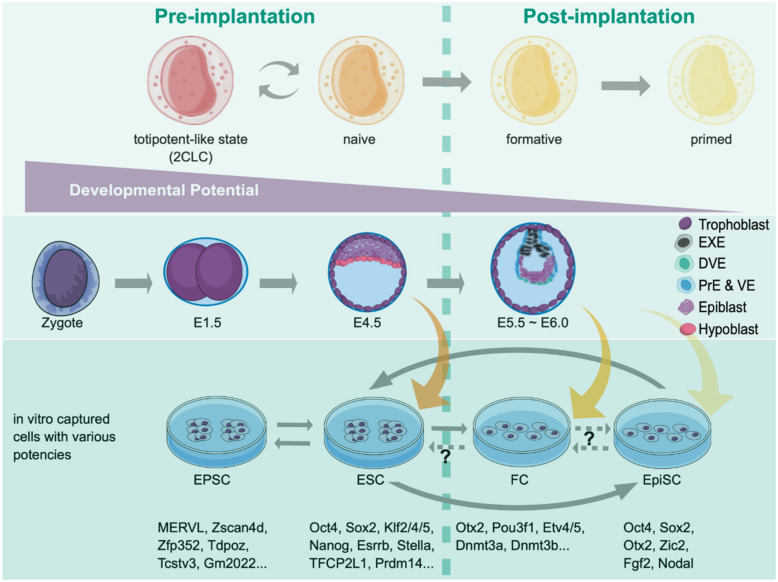
Various pluripotent and totipotent-like states and their corresponding embryonic stages. The developmental continuum of pre- and early postimplantation mouse embryos is depicted, where developmental potency decreases from left to right. Epiblasts captured in vitro correspond to different developing embryos in vivo with the expression of specific markers. The conversion between stages can be achieved by either using various media or introducing specific factor sets, depending on the particular phases of interest. EXE, extraembryonic ectoderm; DVE, distal visceral endoderm; PrE, primitive endoderm; VE, visceral endoderm; Epi, epiblast.

**Figure 3 cells-10-03558-f003:**
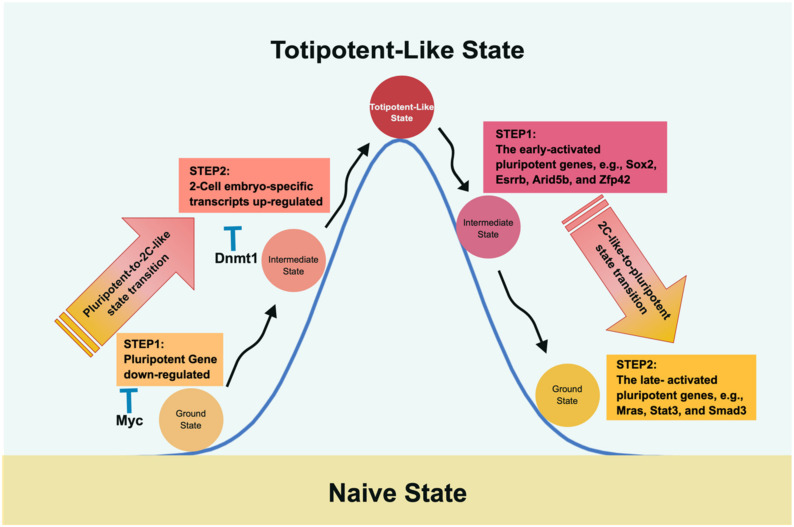
The molecular roadmap of transitioning between pluripotency and 2CLC. Fu et al. depicted the roundtrip transition between pluripotency and 2CLC and vice versa [164,165]. Intriguingly, traveling between pluripotency and 2CLC adopts different mechanisms and, therefore, takes different routes. The path from pluripotency to 2CLC encounters Myc and Dnmt1 as roadblocks that suppress the pluripotent expression level and DNA methylation to impede the transition. In the return route, two gene sets are observed with different temporal expression patterns to restore the pluripotent state; i.e., the late-activated pluripotent genes closely follow the early-activated expression gene set.

**Figure 4 cells-10-03558-f004:**
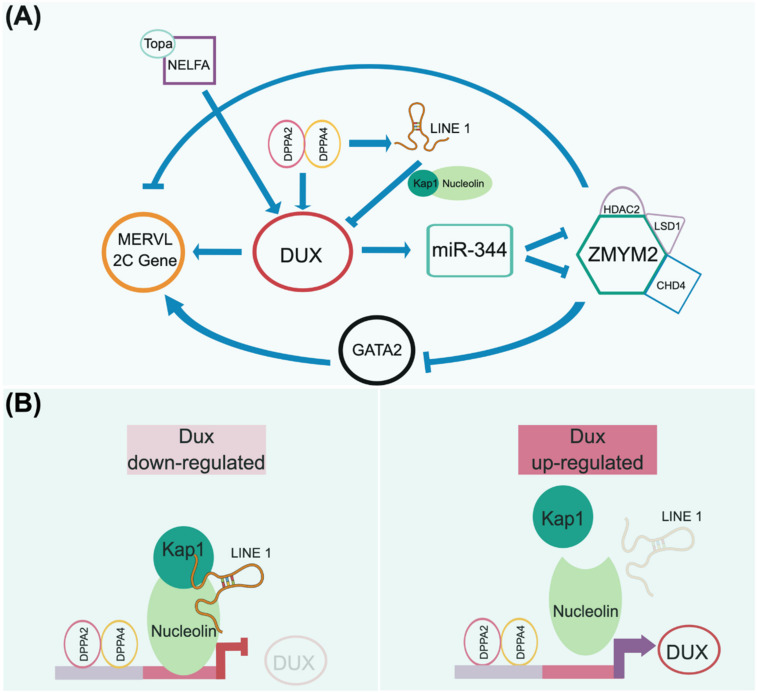
A presumptive totipotency regulatory network. Both maternal and zygotic factors need to coordinate to launch the beginning of development to maintain the totipotent-like state. As 2CLC is currently only observed in mice, we adopt mouse 2CLC as a paradigm to presume its genetic circuitry, as addressed in recently published articles [171,172,173]. (**A**) In the current view, the homeodomain-containing DUX transcription factor plays an essential role in governing the transient 2CLC state. The expression of DUX can be regulated by several maternal factors, e.g., NELFA/Topa and Dppa2/4 complexes. DUX activation triggers downstream events that are uniquely observed in the 2-cell embryo, including the upregulation of MERVL elements. However, the downregulation of DUX is required for further development. Thus, the Dppa2/4 heterodimer complex activates LINE-1 to form a negative feedback loop to repress DUX expression by forming the LINE-1/Kap1/Nucleolin complex (**B**). This regulatory network is consistent with the transient expression of DUX in the 2-cell stage engaging in developmental initiation and, in turn, relaying the totipotent network to the pluripotent circuitry. Notably, other factors may function together with DUX to define the totipotent state, since Dux knockout mice still produce fertile offspring at a sub-Mendelian rate.

## Data Availability

Not applicable.

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
