# Peer review of "Capturing Pluripotency and Beyond"

_cells, 2021, doi:10.3390/cells10123558_

Round 1

Reviewer 1 Report

This article talks about fundamental topics in the stem cell field, about the study of the pluripotent and totipotent state of stem cells, reviews the mechanisms of preimplantation embryonic development in mice and compares them with human embryonic development, and provides a summary study of the mechanisms of stem cell fate transition.

1. The field of stem cells, a hotspot and frontier of medical research, has developed rapidly, especially in recent years, yet the references in this paper are somewhat outdated. A total of 141 references are cited in this article, of which only 11 are from the last three years.

2. What are the innovations and strengths of this article's research compared to previous reviews? Please clarify this points for readers. 

3. Rodent and mammalian growth and development are very different, and due to ethical constraints human embryos have been less studied, and it is not fully explained what the latest advances in human ESC are, what the difficult problems are and where potential solutions lie.

Author Response

  1. The field of stem cells, a hotspot and frontier of medical research, has developed rapidly, especially in recent years, yet the references in this paper are somewhat outdated. A total of 141 references are cited in this article, of which only 11 are from the last three years.

Response:

The present manuscript aims to give audiences a broad spectrum of progress in stem cell biology. Unlike the other excellent reviews elsewhere, which focus only on the latest years’ progress, we would like to let the readers, especially the beginners, have a historical view by addressing the critical issues at each era during the progressively advancing stem cell field. To that end, we concisely present the pivotal issues along the time axis to string with and sketch up the outline of contemporary stem cell biology from Sir Martin Evan’s Nobel work to the recent 2CLC and EPSC findings. In that vein, the manuscript has to cite the corresponding references to the discussed issues instead of bringing in all the current works. Regrettably, several recent outstanding articles cannot be included to discuss in the present manuscript. Yet, to hold the integrity of our original intention for a broader readership, we apologize for those excellent works which cannot be included in this concise review article.

  1. What are the innovations and strengths of this article's research compared to previous reviews? Please clarify this points for readers. 

Response:

Please see the first response.

  1. Rodent and mammalian growth and development are very different, and due to ethical constraints human embryos have been less studied, and it is not fully explained what the latest advances in human ESC are, what the difficult problems are and where potential solutions lie.

Response:

Regarding the human ESC parts, we have intensively discussed the recent debatable issue regarding the existence of the hESC naïve state and the caveat of converting hPSC to naïve-like hPSC from the works of Austin Smith, Jacob Hanna, and Rudolf Jaenisch labs. Here, we stayed at the preimplantation stage since it will be earlier to compare with the mouse counterpart at this stage. Indeed, we are aware of the essential differences regarding the architecture of human postimplantation embryos, displaying a disc-shaped epiblast instead of a cylinder configuration, as seen in rodents. Furthermore, although the hPSC derived from the ICM of the human preimplantation embryo is equivalent to the rodent epiblast harvested after implantation, the difference between humans and rodents in pluripotency has been recognized in the preimplantation genetic network as addressed in the manuscript.

The critical issue regarding the disc-shaped primate embryo, including humans, will be the germ cell fate determination that the Surani group has discussed and significantly contributed. However, the present manuscript focuses on “pluripotency and beyond” instead of the lineage specification. We thereof do not make an effort to discuss the significant differences after the postimplantation event between primates and rodents.

Thank you very much for bringing up this important issue, which I have included to contribute to my next writing list.

Reviewer 2 Report

This manuscript gives concise information about capturing pluripotency but I have no idea whether this article gives novel point-of-view or cutting-edge information to the readers in this research field. Minor points are below.

  1. Please revise the red-point remained in 5. Capturing Totipotency, 5.4 The 2C-like network, 6. Conclusion.
  2. In Figure 3, reading the text in the textbox of STEP 1, 2 is not comfortable. The margin should be expanded to improve readability.
  3. In the legend of Figure 1, there is an explanation about (A) and (B). However, there is no (A) and (B) in the figure.
  4. On page 9 3.6 part and page 12 5 part, "As aforementioned" is repeatedly used. Please re-write with appropriate expression.
  5. In Figure 3 (STEP1. Pluripotent genes), soc2 may be replaced with sox2.

Author Response

This manuscript gives concise information about capturing pluripotency, but I have no idea whether this article gives novel point-of-view or cutting-edge information to the readers in this research field.

Response:

The present manuscript aims to give audiences a broad spectrum of progress in stem cell biology. Unlike the other excellent reviews elsewhere, which focus only on the latest years’ progress, we would like to let the readers, especially the beginners, have a historical view by addressing the critical issues at each era during the progressively advancing stem cell field. To that end, we concisely present the pivotal issues along the time axis to string with and sketch up the outline of contemporary stem cell biology from Sir Martin Evan’s Nobel work to the recent 2CLC and EPSC findings. In that vein, the manuscript has to cite the corresponding references to the discussed issues instead of bringing in all the current works. Regrettably, several recent outstanding articles cannot be included to discuss in the present manuscript. Yet, to hold the integrity of our original intention for a broader readership, we apologize for those excellent works which cannot be included in this concise review article.

Minor points are below.

 Please revise the red-point remaining in 5. Capturing Totipotency, 5.4 The 2C-like network, 6. Conclusion.

  1. In Figure 3, reading the text in the textbox of STEP 1, 2 is not comfortable. The margin should be expanded to improve readability.

Response: Figure 3 does not have “STEP 1, 2” in the text.

  1. In the legend of Figure 1, there is an explanation about (A) and (B). However, there is no (A) and (B) in the figure.

Response: Figure 1 does have (A), (B), & (C) labeled.

  1. On page 9 3.6 part and page 12 5 part, "As aforementioned" is repeatedly used. Please re-write with appropriate expression.

Response: Changed accordingly

  1. In Figure 3 (STEP1. Pluripotent genes), soc2 may be replaced with sox2.

Response: In Figure 3, there is no “soc2” in the content of the text.

Reviewer 3 Report

In my opinion authors wrote really good review with impressive synthesis of published data. I have only one comment. What about induced pluripotent stem cells? How pluripotent or what type of pluripotency they acquire in terms of prime, naive etc pluripotency.

Author Response

In my opinion authors wrote really good review with impressive synthesis of published data. I have only one comment. What about induced pluripotent stem cells? How pluripotent or what type of pluripotency they acquire in terms of prime, naive etc pluripotency.

Response:

Thank you for your comments.

Regarding the mechanism of induced pluripotent stem cells, I have submitted a review-type article entitled "Somatic Reprogramming~ above and beyond pluripotency" in the accompanying manuscript, which is currently under review. Both manuscripts shall help readers understand the current view of pluripotency mechanism and route-taken choice during embryogenesis and somatic reprogramming.

As for the critical questions you have raised, the current reprogramming field mainly focuses on the completeness of somatic reprogramming; i.e., most articles only addressed whether the highly differentiated somatic cells reach the pluripotent state as their ESC counterparts. If not, it will be recognized as a partially reprogrammed cell. In that vein, somatic reprogramming will get primed and naïve states in humans and mice, respectively. However, the recent advances have allowed the pluripotent cells to advance to a two-cell stage-like cell (2CLC), a totipotency-like state, in both iPSC and ESC.

Further, theoretically, those partially reprogrammed cells can be captured at different reprogram stages if a suitable environment is provided. Therefore, several reports adopted this concept to produce a progenitor-like cell using an interrupted reprogramming approach. However, considering the unique property of progenitor cells' self-renewal and lineage-restriction, the induced progenitor cell for a specific cell lineage perhaps is a safer and more effective approach than using iPSC in clinical settings. I have addressed these interesting issues in the accompanying paper. Again, I thank you very much for bringing up these important questions.

Round 2

Reviewer 1 Report

The current progress should be cited and reflected in high degree in a review article even if that is a text book of new version.   

Author Response

Thank you very much for your thoughtful consideration. I have included the recent progress as shown in the text labeled in red (page 8, 9 &11).

Reviewer 2 Report

I requested several minor points as below but some responses from the authors are not acceptable.

 In Figure 3 (of Page 13), reading the text in the textbox of STEP 1, 2 is not comfortable. The margin should be expanded to improve readability.

Your response: Figure 3 does not have “STEP 1, 2” in the text. --> not in the text, in the figure itself (Page 13).

In Figure 3 (STEP1. Pluripotent genes), soc2 may be replaced with sox2.

Your response: In Figure 3, there is no “soc2” in the content of the text. --> not in the text, in the figure itself (Page 13).

Author Response

I requested several minor points as below but some responses from the authors are not acceptable.

  1. In Figure 3 (of Page 13), reading the text in the textbox of STEP 1, 2 is not comfortable. The margin should be expanded to improve readability.

Your response: Figure 3 does not have “STEP 1, 2” in the text. --> not in the text, in the figure itself (Page 13).

  1. In Figure 3 (STEP1. Pluripotent genes), soc2 may be replaced with sox2.

Your response: In Figure 3, there is no “soc2” in the content of the text. --> not in the text, in the figure itself (Page 13).

Response:

Yes, I found and corrected them all as you suggested. Again, I apologize for my neglect and carelessness.